# Coverage and Representativeness of Passive Surveillance Components for Cattle and Swine in The Netherlands

**DOI:** 10.3390/ani12233344

**Published:** 2022-11-29

**Authors:** Imke Vredenberg, Gerdien van Schaik, Wim H. M. van der Poel, Arjan Stegeman

**Affiliations:** 1Department of Farm Animal Health, Faculty of Veterinary Medicine, Utrecht University, 3584 CL Utrecht, The Netherlands; 2Royal GD, 7400 AA Deventer, The Netherlands; 3Wageningen Bioveterinary Research, 8221 RA Lelystad, The Netherlands

**Keywords:** animal health surveillance, coverage, representativeness, helpdesk contacts, postmortem submissions

## Abstract

**Simple Summary:**

To protect animal and human health, it is important to detect disease early and prevent spreading. Other common aims of animal health surveillance are disease monitoring and proving freedom of disease. In a good surveillance system, representative information of farmers and veterinarians from the entire country should be collected. Our study focused on the validity of the Dutch passive surveillance components based on veterinary helpdesk and postmortem data. We examined whether data was received from the entire country and represented the swine and cattle sector in the Netherlands. The difference between the region with the highest and lowest contact rate was 12.8 and 6.1 fold for cattle and pigs, respectively; for postmortem, this was 4.0 and 38.3. Part of the differences could be explained by the distance to the postmortem facility and farm density. We found that veterinary practices serving few farms and veterinary practices serving many cattle farms had fewer contacts compared to other practices. Although this study found regions and practices with lower-than-expected contacts and postmortems, information was obtained from all regions and most veterinarians. The results can be used to improve the coverage and representativeness by an increased focus on specific regions, farms or veterinary practices.

**Abstract:**

Common aims of animal health surveillance systems are the timely detection of emerging diseases and health status monitoring. This study aimed to evaluate the coverage and representativeness of passive surveillance components for cattle and swine in the Netherlands from 2015–2019. The passive surveillance components consisted of a telephone helpdesk for veterinary advice and diagnostic and postmortem facilities. Spatial analysis showed heterogeneity (range in RR = 0.26–5.37) of participation across the Netherlands. Generalized linear mixed models showed that distance to the diagnostic facility and farm density were associated with the number of contacts of farmers with the helpdesk and postmortem examination. The contact rate of veterinary practices was associated with their number of clients, ranging in RR from 0.39 to 1.59. We concluded that the evaluation indicated differences in coverage of the passive surveillance components across regions, farms and veterinary practices. Due to the absence of emerging infections in the study period, we were unable to estimate the consequences of the observed differences for the early detection of disease. Nevertheless, regions and veterinary practices with low participation in passive surveillance might be a risk for early detection, and consequently, further understanding of the motivation to participate in passive surveillance components is needed.

## 1. Introduction

A national animal health surveillance system (AHSS) has a number of important aims, among which timely detection of emerging diseases and monitoring of the health status of the animal population are most common. The system should give insight into the prevalence of endemic diseases and prevent major outbreaks of (exotic) diseases by allowing a timely reaction to anomalies [1]. Surveillance is characterized by the attached intervention plan aiming to control disease, different from monitoring, where intervention is normally absent [2]. In addition to the main objectives to maintain and improve the health status of the animal population, the risks of zoonotic disease transmission make AHSS also important for food safety and human health [3]. In addition, the AHSS’s objective to prove disease-free status is essential for the international trade of animals and their products. A good working AHS, therefore, is not only important for animal and human health but also of economic importance [4]. 

In April 2021, the new European Animal health law came into effect [5] and replaced the Community Animal Health Policy, which included multiple legislations and guidelines. The Animal health law encompasses requirements for all EU countries to have an AHSS in place to prevent and control animal disease and detect emerging diseases [6]. The legislation includes a list of diseases, including specifications on how EU countries have to act when one of these diseases is or becomes present in their country [7]. Although prescribed criteria have to be met, countries can adapt the framework to best fit their current health status, farm industry and infrastructure [5]. Therefore, countries have different surveillance systems for different diseases, and due to country differences (e.g., livestock species and trade), there is not a single best general AHSS for all countries [8]. Systematic evaluations are needed to determine the functionality and quality of these AHSS. 

AHSS usually has different passive and active components for data collection. In a passive approach, data comes from the field at the initiative of farmers, veterinarians and other stakeholders [2]. Passive surveillance may result in the underrepresentation of specific groups or geographic regions. An active approach means that data collection is initiated by the AHSS. This offers the possibility to reduce the bias of underrepresentation mentioned above. Peyre et al. developed the EVA tool, a framework for the evaluation of AHSS [4]. This framework includes a list of attributes that can be integrated into an AHSS evaluation. Representativeness and coverage are included in this attributes list and are defined as “the extent to which the features of the population of interest are reflected by the population included in the surveillance activity“ and “the proportion of the population that was included in the surveillance”. Features of representativeness may include herd size, production type, age, sex or geographical location and time of sampling. Evaluation of these attributes is an important step to investigate if groups or regions are underrepresented, increasing the risks of not detecting disease outbreaks in a timely manner. 

The Netherlands is an important importing and exporting country with a dense animal and human population. International trade combined with these high densities results in an increased risk of outbreaks with major consequences, making a well-functioning AHSS important. In the Netherlands, the voluntary national AHSS system is commissioned by Royal GD (Gezondheidsdienst voor Dieren) [9] by the livestock industry and the Ministry of Agriculture. The AHSS consists of surveillance components to actively (e.g., disease prevalence, trend analyses of census data) and passively (e.g., veterinary helpdesk, postmortem examination) collect data about the animal health status [10]. These components aim to follow trends and detect introductions and outbreaks of unknown or exotic diseases. When one or a combination of component(s) results in suspicion of a notifiable disease, the veterinary authority (i.e., the Dutch Food Safety Authority) has to be contacted immediately to plan and start following-up investigations and interventions when applicable. Passive surveillance holds a potential risk of over or underrepresentation of specific groups or regions. Scientific evaluation of representativeness and geographical coverage of the data has not been performed previously. Because passive surveillance is an important component of disease control in many, if not all, countries, such evaluation is highly relevant to other countries. 

The aim of this study was to evaluate the representativeness and coverage of passively collected data from the veterinary helpdesk and postmortem examination in the Dutch cattle and pig sector. Insight into the representativeness of passively collected data is important to find over and underrepresented groups and regions, which can cause bias in the data, potentially resulting in a wrong interpretation of the situation within a country. Poor representation of an AHSS can result in missing signals and, therefore, negatively influence the timeliness of disease detection with possible health risks for the animal population in the country itself but also for the populations in neighboring countries and trading partners.

## 2. Materials and Methods

### 2.1. Study Population

This study focused on the Dutch cattle and pig population. Although both species are important for Dutch animal production, structures of these sectors differ significantly. The cattle sector consists of more farms with smaller numbers of animals than the pig sector and is more equally distributed across the country. Service by veterinary practices in cattle is more locally organized compared to the pig sector. 

The number of farm animals, their locations and their movements are recorded by the Dutch identification and registration system (I&R). This database was used to determine the number of cattle and pig farms each year. The total number of cattle was available from the national identification and registration system (I&R). Unlike cattle, pigs are not individually registered in I&R, and therefore, their total number was not available. The information was collected for each year from 2015 to 2019. Within this period, there were no notifiable disease outbreaks in the Netherlands in the cattle population or in the pig population.

### 2.2. Data Collection

Data consisted of information from two surveillance components of GD, the veterinary helpdesk and postmortem examination. The veterinary helpdesk can be contacted by veterinarians and farmers, either by phone or e-mail, with all sorts of questions related to animal health. The overall purpose of the system of the helpdesk is to identify early signs of emerging diseases. However, the reason for veterinarians and farmers to contact the helpdesk is often to obtain help with an animal health problem that they encounter. The employees of the helpdesk are trained veterinarians. The phone calls and e-mails are registered in a system from which the data used in this study were obtained. In 2019 the registration system changed. However, similar information was registered. Each week, the aggregated information is discussed and analyzed to timely detect anomalies and take follow-up actions when needed. The following information is registered and was available for this study; farm and veterinary practice ID and location, animal species (cattle, pig), animal type (e.g., milking cows, calves, sows), categorized reasons for contact (e.g., clinical signs, disease, management), anonymous contact (yes/no), suspicion of a notifiable disease (yes/no), new or emerging disease (yes/no), change in endemic situation (yes/no), date and how the contact took place (e.g., phone, e-mail). 

Postmortem examination is another surveillance component at GD for early detection of emerging diseases as well as a diagnostic tool that provides information on cause of death to the farmer and veterinarian. This surveillance component is subsidized, and farmers thus pay a fairly low rate. When animals are registered for postmortem examination, GD will send a vehicle to collect the animals, after which necropsy will take place. The costs of transport and necropsy are independent of the distance of the farm to GD. The number of submitted animals and the results are stored in a database. For our study, we used date of submission, farm ID and location, animal species and the number of animals per submission. 

Additional to the postmortem examination and helpdesk data, two smaller datasets were available: information on veterinary practices about the number of farms that they attend (each farm in the Netherlands has to be linked to a veterinary practice), the location of the practice and information of the total number of farms per year per postal code area. 

### 2.3. Location Definition

The previously described data contained the location of the farms and veterinary practices. For privacy reasons, the data was anonymized, and locations were categorized by the first two numbers of the Dutch postal code system (PC2) (Figure 1). This system is based on the human population density. Population-dense regions within one PC2, like cities, have a smaller total surface (km^2^) compared to more rural areas, having more surface within one PC2 area. The use of this system divides the Netherlands into 90 areas.

### 2.4. Data Inclusion

A number of analyses were performed to obtain insight into the representativeness and coverage of the helpdesk and postmortem examination data. For each analysis, a different subset of the data was used. The helpdesk data contained several variables with missing values. The inclusion criteria for each analysis were based on the availability of critical variables for that analysis to be performed (i.e., Herd ID, Vet ID, registration of clinical signs). When those values were missing, the record was removed. Table 1 shows the inclusion criteria per analysis.

### 2.5. Reported Clinical Signs

The dataset of the veterinary helpdesk included specifications on the clinical signs addressed in the phone calls or e-mails, although this was not registered for all records. As multiple clinical signs can be registered per contact, the function “grepl” of the R package “base” was used to subtract all different categories, after which the frequency per year was counted. In total, 22 and 53 problem categories were identified for cattle and pigs, respectively. These were further combined based on organ system leading to 14 categories for cattle and 15 for pigs (Table A1). Bar and pie plots were made to compare the frequency of the different categories to see if there were any patterns in the type or frequency of reported clinical signs between years. The R-package “ggplot2” and “plotly” were used to make the figures. The R-version used for this study was 4.0.2. 

### 2.6. Space-Time Analysis 

The first analysis was a space-time cluster analysis performed in SaTScanTM (v9.6.1 64-bit) [12]. This program can perform spatial cluster analysis to detect significant clustering of data over spatial areas within a specified timeframe [13]. The program uses cylindric shapes in which the base covers the space component, and the height of the cylinder covers the time. The program determines the best-fitting cylinder, followed by the second best and so on. 

SaTScanTM needs a case, population and geographic file to run the analysis. The postmortem examination data and helpdesk data for pigs and cattle were used to perform four space-time analyses, one for each dataset per species. Because location was essential for the analysis, records with missing values for PC2 were excluded. The case file contained the number of contacts or postmortem examinations, the population file, the total population per location, and the geographic file contained the longitude and latitude coordinates of the centroids of the PC2 areas.

Based on the three input files, the program determines the expected value, the likelihood and relative risk (*RR*) for a cluster. Equation (1) shows how the expected value *E[c]* was calculated with *C* as the total number of cases in the population, *P* as the total population and *p* the population within i, where i can be a single region or cluster. The *RR* can be calculated by Equation (2) where *c* is the total number of cases in region i, *C* is the total number of cases and *e* is the expected value calculated by Equation (1).
(1)$$E\left[c\right]=p\frac{C}{P}$$
(2)$$RR=\frac{c/e}{\left(C-c\right)/\left(C-e\right)}\;$$
(3)$$\frac{L\left(z\right)}{L_{0}}=\frac{{\left(\left.\frac{c}{E\left[c\right]}\right)\right.}^{c}{\left(\frac{C-c}{C-E\left[c\right]}\right)}^{C-c}}{{\left(\frac{C}{E\left(T\right)}\right)}^{C}}$$

SatScan uses a likelihood ratio test to determine if the null hypothesis (H_0_), the *RR* is equal in all areas, is true, or if the alternative hypothesis (H_A_) is true, there are areas with a different *RR*. Equation (3) shows how this was calculated by the program. *L*(*z*) is the likelihood for the cluster, *L_0_* is the likelihood under H_0_, *c* is the number of cases in i, *C* is the total number of cases, *E*[*c*] is the expected number of cases, and *E*[*T*] is the expected number of cases in i over time. When the outcome is different from 1, H_0_ has to be rejected. In that case, there are clusters that have an increased or decreased *RR*. Although SatScan also gives non-significant clusters, only significant clusters (*p* < 0.01) are shown in the result section.

Table 2 shows the SatScan settings for the analysis that differed from the default settings of the program. We used space-time analysis to be able to look retrospectively in space and time simultaneously for high and low clusters with the default use of the Poisson distribution. This cluster setting was chosen because, for surveillance purposes, clusters with low RRs are interesting to show gaps in representativeness; however, clusters with high RRs show where there is overrepresentation, both of which can lead to wrong conclusions about disease status. One cluster could obtain a maximum of 15% of the population at risk to prevent single clusters covering a large part of the country. The chosen maximum of 15% made the clusters more precise and more informative in terms of representativeness. Because the 15% was an arbitrary choice, a sensitivity analysis was performed using a maximum of 20% and a maximum of 10%. Using a maximum of 10% resulted in a partitioning of larger clusters into several smaller clusters in the same areas. The opposite occurred using 20% as maximum, where large clusters became even larger. The largest difference in RR was found in one cluster for which the RR increased from 2.14 (max. cluster size 15% and 20%) to 3.64 (max. cluster size 10%). Although there were some changes, most differences were small and did not result in different conclusions. The minimum and maximum time periods of the clusters were set at 2 and 4 years, respectively. Clusters of only one year were prevented with this setting. Clusters existing for one year were not included, as the aim of the study was representativeness of an ongoing surveillance system. Clusters of one year could exist due to chance, as there are regions with only a few farms. Setting the maximum temporal cluster size beyond 4 years was analytically not possible. 

### 2.7. GLMM Association Distance GD, Farm Density, DAP Size with Contact

Three associations were examined using GLMM. The first two associations were the distance of the farm to GD and the farm density compared to postmortem examination and helpdesk contact. Farm distance was expected not to be related to the number of submissions for postmortem examinations nor helpdesk contact because cost and availability were independent of the distance, but it was tested to confirm this hypothesis. The association was also examined for the helpdesk contact, although there was no association expected. The distance to GD was determined as the linear distance from the centroid of the PC2 area to the GD facility and divided by ten, resulting in a per 10 km unit. Farm density was also added to the model and was defined as the number of farms divided by the surface in km^2^ of the PC2 area. The farm density was divided into 5 categories, each including 20% of the data points. The first group had the lowest density and the last group the highest. 

Four GLMM models were run to determine the effect of the distance to GD and the farm density on the outcome variables contact and postmortem examination in both pigs and cattle. In all models, the variable ‘year’ was forced into the models. Because of the skewness of the data, three models were run with a negative binomial distribution. For the model on ‘helpdesk contact’ for pigs, a Poisson distribution was used. The negative binomial model did not fit the number of contacts for pigs. In all models, an offset with the number of farms within the PC2 area, and a random intercept with the PC2, were used to account for differences in the numbers of farms and multiple measurements per PC2 area.

The third analysis focused on the frequency of contacts and the size of the veterinary practice. Size was defined as the total number of relations (farms) registered to the veterinary practice for the specified animal species. The hypothesis was that larger practices had more expertise and were, therefore, less likely to contact the helpdesk. In that case, information about animal health and possible outbreaks can be missed or detected later. The size of veterinary practices was categorized (Table 3). The lowest categories represent veterinary practices assumed not to have pigs or cattle as core business. This was also the motivation to use category 2 as reference category in the models, instead of category 1, for both species. The highest category contained the top 5% biggest veterinary practices. The remaining veterinary practices were divided into three groups using the AIC for optimization. A subset of the helpdesk data was used with availability of veterinary practice ID as inclusion criteria. 

Two models were run to determine the association between the size of the veterinary practice on contact. The dependent variable was the frequency of contact. The independent variables were veterinary practice size and year. The number of relations was added as an offset to the models. For both models, the negative binomial distribution was used due to skewness of the data. For all GLMM models using the negative binomial distribution, the “Mass” package of R was used.

## 3. Results

### 3.1. Descriptive Statistics

In the period of 2015 to 2019, the veterinary helpdesk received a total of 19,615 and 6047 phone calls and e-mails for cattle and pigs, respectively (Table 4 and Table 5). In the last two years, there was a decrease in the number of contacts for cattle. In contrast, for pigs, there was a slight increase from 2016. The location and ID of the farm, veterinary practice or both were registered for 89% and 81% of the cattle and pig contacts, respectively. For both species, there was an increase in the registration of the location and IDs in 2019, with the largest increase for pig farm ID registration. In total, 62% and 46% of the contact was about clinical signs for cattle and pigs, respectively. For cattle, this was relatively stable over the years, while for pigs, this slightly increased. Additionally, the helpdesk registered if there was a suspicion of notifiable or zoonotic risk. For both species, there was a slight increase in reporting of a suspicion of a notifiable disease in 2019. For perceived zoonotic risks, the percentage slightly decreased. The number of postmortem examinations varied between 2370 in 2019 and 3003 in 2016, with no clear trend. The number of farms and veterinary practices for pigs and cattle decreased, but the number of cattle did increase (Table 6). This indicates that the average number of animals per farm has increased over the last few years.

### 3.2. Reported Clinical Signs

When farmers and veterinarians reported clinical signs to the veterinary helpdesk, these were registered in the system. For cattle, in 62% of the contacts, clinical signs were reported; for pigs, this was 46% (Table 4 and Table 5). Figure 2 shows the percentages of reported signs across the various categories for cattle and pigs. Miscellaneous signs (20.6%) were the largest group for cattle, followed by udder-related signs (20%), gastro-intestinal signs (11%) and mortality (10.4%). For pigs, the largest groups were stragglers (15.5%), lameness (13.6%) and general advice (11.7%). While in cattle, udder-related signs were clearly the largest single clinical problem, the frequency was more evenly spread over the first seven categories in pigs, which were all above 10%. When comparing the categories over time, for pigs, there was an increase in the general advice category in 2019 (Figure A1).

### 3.3. Spatial Farm Distribution

The cattle farms in the Netherlands are mostly concentrated in the center, north and east part of the country. The number of farms per PC2 area lies between 7 and 1309 farms (Figure 3A). Figure 3B,C show the spatial distribution of the number of postmortem examinations and help desk contacts per PC2 area. The figures show an overlap between the farm distribution and the distribution for postmortem examination and contact.

For pigs, most farms are concentrated in the south and east of the Netherlands. The densest PC2 area consists of 601 pig farms, accounting for almost 10% of all pig farms. Even though the number of farms for cattle and pigs decreased, the distribution of the farms over the separate years did not change.

### 3.4. Space-Time Cluster Analysis

Figure 4 shows the spatial results of four space-time cluster analyses for cattle and pigs. All these clusters were significantly associated with contact or postmortem submissions (*p* < 0.01). The numeric results corresponding to the clusters can be found in Table A2. In the table, the population, case and expected numbers are summed over the years of existence of the cluster. For the postmortem examination of cattle, there were six significant clusters identified (Figure 4A). Three of these clusters (1, 5 and 6) had a higher submission rate than was expected (RR = 2.14, RR = 1.41 and RR = 1.17). Clusters 1 and 5 were present for a maximum period of four years. The three clusters were located above each other from north to south. Clusters 2, 3 and 4 had significantly lower submission rates than expected (RR = 0.56, RR = 0.53 and RR = 0.54), of which 2 and 4 were present for four years. Of all clusters, cluster 3 was the largest cluster containing 22 PC2 areas. 

The analysis of helpdesk contacts for cattle showed a different pattern compared to the postmortem examination data (Figure 4B). Clusters 2, 5, 6, 7 and 8 had less contact than expected and are located in the middle of the country (RR = 0.57, RR = 0.67, RR = 0.77 and RR = 0.53). Cluster 8 also had fewer contacts than expected but was located more to the south (RR = 0.28). Clusters 1, 3, 4, 9, 10, 11 and 12, with more contacts than expected, are located in the north and south (RR = 3.60, RR = 1.53, RR = 1.86, RR = 1.60, RR = 1.49, RR = 1.56 and RR = 1.74). Most clusters (7 of 12) did exist for the maximum period of four years (Table A2). 

The analysis of the pig data showed a more comparable pattern of the clusters between postmortem examination and helpdesk contacts (Figure 4C,D). In general, there are large amounts of PC2 areas in the low clusters (46 and 51 PC2 areas), and they are mostly in the north and west of the country, where there are few pig farms. For postmortem examination, these are clusters 2, 5 and 8 (RR = 0.47, RR = 0.26 and RR = 0.14), and for contacts with the helpdesk 4 and 7 (RR = 0.42 and RR = 1.63). Most clusters existed for three or four years. The largest difference between postmortem and contacts can be found in the south-central area (Figure 4C, clusters 1 and 9 of the postmortem examination (RR = 5.37 and RR = 1.33) and Figure 4D, cluster 2 for contacts (RR = 0.42)). For postmortem examination, this region was marked as a high cluster, while for contacts, the area is a low cluster. When comparing Figure 3D with Figure 4C,D, most of the high clusters are within dense farm regions, and low clusters are in low-density regions for pigs. For cattle, this association is not visible.

### 3.5. Association Distance GD and Farm Density

The association of distance to the GD site and the farm density in the area with contact with the helpdesk and postmortem examination was estimated by three negative binomial models and one Poisson model (Table 7). In all models, the association of contact or postmortem examination between farm density and distance to GD was significant for at least one category and the association was positive. Farm density was negatively associated with contact and postmortem examination, meaning fewer contacts or submitted animals in high-density areas compared with the moderate group. Although not all years were significant in all models, in pigs, an increasing trend can be seen. For cattle, no trends were found.

### 3.6. Association Size Veterinary Practices 

Figure 5 shows the relation between the number of registered farms to a specific veterinary practice for the specific species and how often they contacted the veterinary helpdesk. For both species, the Figure shows that there was more contact when the veterinary practice had more relations. This was stable over the years, although the curve for pigs in 2015 bends slightly in the end, in contrast to the other years. 

Two negative binomial models, with the number of relations as offset, were run to determine the relationship between the frequency of contact and the number of relations of a vet for that species. For cattle, the lowest (<15) and highest relation categories (300<) were significantly different from the reference group (15–80), with a factor of 0.52 and 0.71, respectively (Table 8). There was no trend between the years. Rates varied from 0.81 to 0.99, of which only the year 2018 was significantly lower than in 2015. For pigs, there was an increasing trend in rate (0.72–1.47) over the years compared to the reference (2015). All years, except 2017, were significantly different from 2015, while 2018 and 2019 were higher and 2016 were lower. Except for the largest category (IRR = 1.05), all other categories did differ from the smaller reference group (10–30). Where the smallest category had less contact (0.39), the small (31–60) and larger (61–100) categories had more contact compared to the reference (IRR = 1.37, IRR = 1.59).

## 4. Discussion

We studied the coverage and representativeness of passively collected data from the veterinary helpdesk and postmortem examinations. These are components in the Dutch cattle and pig health surveillance system aiming for the early detection of emerging diseases. The characteristics of the monitoring data were examined by descriptive statistics, spatial-temporal analyses and GLMM models. Coverage and representativeness can indicate the effectiveness of surveillance components and were defined as “the proportion of the population that was included in the surveillance” and “the extent to which the features of the population of interest are reflected by the population included in the surveillance activity”, respectively [5]. 

The helpdesk registered 19,615 contacts for cattle over five years, on average covering 3.5% of the total cattle farms and 55.6% of the veterinary practices per year. For pigs, 6047 contacts were registered, covering on average 2.7% of the pig farms and 33.0% of the veterinary practices per year. Over the five years, 2.5% of the cattle veterinary practices with more than 15 relations and 2.2% of the pig veterinary practices with more than 10 relations had no contact with the helpdesk. There were 13,219 postmortem examinations for cattle and 12,300 postmortems for pigs, on average covering 4.5% and 10.2% of the farms per year, respectively. Overall, farmers further away from GD had more contacts and submitted more animals for postmortem examination. We saw the same result for farmers in the lowest farm-dense areas and cattle farmers located in lower farm-dense areas. We did not have an explanation for these results, and further research, such as interviews with farmers and veterinarians, is needed for clarification. Pig farmers located in the highest farm-dense areas had a lower rate of postmortem examination. The number of contacts of veterinary practices was associated with the number of clients. For cattle, veterinarians with more than 300 clients had relatively fewer contacts per client compared to the reference group with 16–80 clients. In contrast, veterinarians from practices with 31–100 clients with pigs had more contacts than the reference group with 11–30 clients. 

The proportion of registered clinical signs for cattle was relatively stable, while for pigs, we saw an increase over the years. In both species, clinical signs related to known endemic health problems were most frequently reported. Clinical signs potentially related to emerging diseases (e.g., mortality, respiratory disease, fever) were reported as well. As the main aim of passive surveillance is the early detection of emerging or exotic diseases, it is important that non-specific clinical signs are reported. Late or no reporting of these signs can result in delayed detection of disease outbreaks. In 2019, we saw an increase in suspicions of notifiable disease and a decrease in possible zoonotic risk. Because these changes took place in both species in the same year and there were no national outbreaks known for these species, the change is most likely attributable to the new registration system that was introduced in 2019. “Miscellaneous” was the largest category for cattle. As the registration for cattle lacks a category for general advice (e.g., eradication programs, lab results), which was largest for pigs, this category was expected to contain similar topics.

We saw a decreasing trend in the number of contacts for cattle. The decreasing trend in absolute numbers of helpdesk contact for cattle may be explained by the decrease in the number of farms. In the Netherlands, there is a trend that the number of farms decreases while the remaining farms increase in herd size. In contrast with cattle, we saw an increase in the number of contacts for pigs, while in this sector, a decrease in farms is also observed. Nevertheless, the decrease for cattle and increase for pigs, cattle farmers and veterinarians still had a higher percentage of contacts than pig farmers and veterinarians. In contrast, for postmortem examinations, the proportion of farms submitting pigs was higher than for cattle. The discontinued farms seem to be randomly spread over the country. Thus, the spatial distribution of farms remained similar over years. Therefore, differences in spatial distribution nor decreasing herd numbers between years explained differences in coverage or representativeness.

In general, the clusters of the spatiotemporal analysis matched well with the farm density, but some specific clusters did not match. For cattle, such a cluster was located in the middle of the country and was underrepresented for contacts but overrepresented for postmortem examination. This can likely be explained by the distribution of different cattle farm types in the Netherlands. Veal farms are mostly concentrated in this specific area, whereas dairy farms are more equally spread across the country. The hypothesis is that veal farmers are more inclined to submit their calves for postmortem examination, while dairy farms are probably more selective or prefer to first contact the helpdesk before submitting cattle. In the midsouth of the country was a comparable situation for pigs. We do not have an explanation for this observed inconsistency. Although excluding records of the helpdesk with missing farm IDs or locations may have introduced bias, we expect this to be limited when the information was missing randomly and was not related to the variables of interest.

The number of contacts of veterinary practices was partly related to their number of clients. The veterinary practices for both species with the least clients had significantly fewer contacts per client compared to the reference, which was expected. It is hypothesized that these veterinary practices serve smallholder farms (e.g., private persons and petting farms). Therefore, this category was expected to be different from other categories. In cattle, the largest veterinary practices had significantly less contacts with the helpdesk compared to the reference. This group of veterinary practices has more specialized veterinarians resulting in less need for secondary advice and thus less contact with the helpdesk than the smaller veterinary practices. This does not necessarily decrease the likelihood of early detection of emerging diseases as long as they contact the helpdesk when unexplained changes in health status or suspicion of the notifiable disease occur. Otherwise, this will increase the risk of delayed detection of disease outbreaks. In pigs, the largest veterinary practices had no different contact rates, while the two groups of middle-sized practices had more contacts per client compared to the reference group. Overall, there were no indications that the size of the veterinary pig practices would influence the contact rate and thus impact the effectiveness of early detection of emerging diseases.

Although the cluster analysis gave insight into over- and underrepresented geographic clusters, these clusters are not necessarily a risk regarding the early detection of disease. The program SatScan uses the population mean to calculate the predicted value of the areas. As far as known, there are no thresholds for what a population mean should be for passive surveillance systems to detect disease timely. We hypothesized that the mean of the low clusters might be high enough already to detect an emerging disease outbreak if only the signals on emerging diseases are directly reported, and no other disease information is shared. Additionally, fewer contacts and postmortem examinations are expected when the animal population is healthy compared to a diseased population. Formulating a general threshold will therefore be very difficult. In addition to the absence of a threshold, defined clusters can be sensitive to small changes in areas with just a few farms. When one farmer called twice, this already caused a significant cluster. The opposite occurred when there were no contacts by these few farmers, which resulted in a low cluster. In-depth surveys and interviews are needed to further examine differences in the motivation of farmers and veterinarians.

In our study, only farm density was included as an explanatory variable in the model. The animal density was not available and, therefore, not taken into account. In general, farm and animal density will be strongly correlated. In addition, between farm spread of emerging diseases, the farm density is more important than the animal density [14]. Farm density was categorized into five groups. Although this categorization did not always result in the best model based on the AIC compared to using farm density as a continuous variable, categorization was preferred for easier interpretation of the results. Furthermore, our conclusions would be similar. Veterinary practices were categorized by the number of relations (i.e., the number of clients). The smallest category was chosen by expert opinion, assuming this category only served smallholder farms. The other categories were chosen on the model’s AIC. In the Netherlands, there are increasingly large corporations and collaborations between veterinary practices. For this study, data on these corporations and collaborations was not available. Correcting for effect was, therefore, not possible.

Combining the spatial results, there were over- and underrepresented regions, but data (contact or animal submission) is coming from all regions. For pigs, contact with farmers in farm-dense areas could be improved. The same was true for the contacts of the largest veterinary practices for cattle. Although there is some over- and underrepresentation, it is not clear if this really affects the early detection capacity of passive surveillance. For early detection receiving the right signals is of more importance than the amount of contact, but it is difficult to define “right signals”. This will depend on the disease, the frequency of reporting within a certain timeframe and the difficulty of recognizing the disease. When the percentage of postmortem examinations of the total number of farms in the Netherlands (7.6% cattle, 35.1% pigs) was compared with other countries with a similar passive surveillance system, the Netherlands ranked in between Flanders (Belgium) (±8.5% cattle; ±55.1% pigs) and Scotland (±4.7% cattle, ±23.1% pigs) from 2015 to 2019, [15,16,17,18]. Both countries have comparable systems; although Scotland lacks a pick-up service for the carcasses, combined with the larger distances to the facilities, this can have influenced the percentages for postmortem examinations [19]. Flanders has a much higher percentage of postmortem examinations in pigs. Most likely, this is related to the outbreak of African Swine Fever in wild boar in September 2018, which likely influenced the numbers in 2019 [20]. 

In the Netherlands, history already showed the effectiveness of the surveillance as it detected the first signals of an outbreak of bluetongue virus (BTV) in August 2006 and of the Schmallenberg virus in 2011 [21,22]. Regarding BTV, this signal concerned contact with the helpdesk reporting 10 seriously diseased sheep, which were found positive for BTV after blood tests. At the beginning of the Schmallenberg outbreak, the helpdesk received an increased number of reports about production losses and diarrhea. As far as we know, only one comparable study has been conducted on the coverage and representativeness of passive surveillance in Scotland [23]. Their study focused on the diagnostic submissions of pigs, cattle and sheep. Based on a histogram representing the proportion of submitting holdings and the distance to the closest surveillance center, they concluded that the distance to the closest Disease Surveillance Centre did influence the rate of submissions. In our study, we did not find a negative influence of distance on the submission rate. This difference may be caused by the existence of a collection service for carcasses in the Netherlands, while in Scotland, the farmer or vet has to arrange the transport themselves [19,24].

## 5. Conclusions

In conclusion, there were over- and underrepresented regions, but it is difficult to draw conclusions about the effect on the early detection capacity of the passive surveillance components. Most underrepresented regions contained only a small amount of farms, and information (contact or postmortem submissions) was coming from all regions. For early detection of emerging diseases receiving the right signals may be more important than the amount of contact. However, the chance of getting the right signal is probably higher with more contacts. Research on the features of underrepresented regions, especially in pigs, and the large veterinary practices for cattle can improve the representativeness and coverage of the surveillance and, therefore, increase the chance of early detection of emerging diseases. The importance of the heterogeneity between regions, farms and veterinary practices remains unclear. Surveys and interviews are needed to determine the factors or context that motivates farmers and veterinarians to contact the helpdesk or submit animals for postmortems. The yearly analysis will help keep track of changes in the representativeness, coverage and, with that, the early detection capacity of the passive surveillance, and it allows for actions when needed. 

## Figures and Tables

**Figure 1 animals-12-03344-f001:**
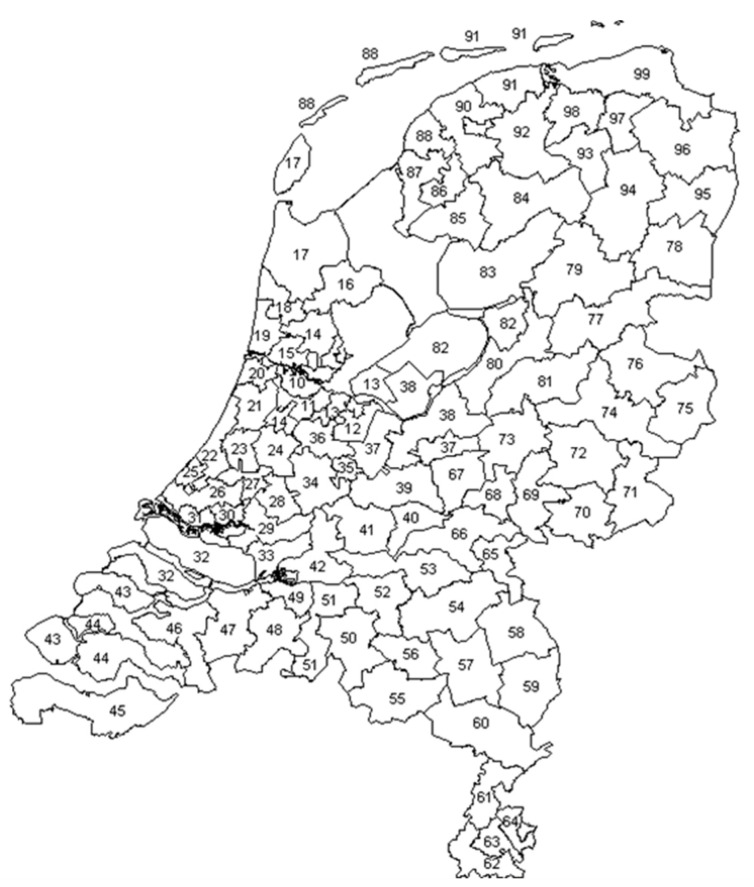
Distribution of the Netherlands based on the first two numbers of the postal code (PC2) [11].

**Figure 2 animals-12-03344-f002:**
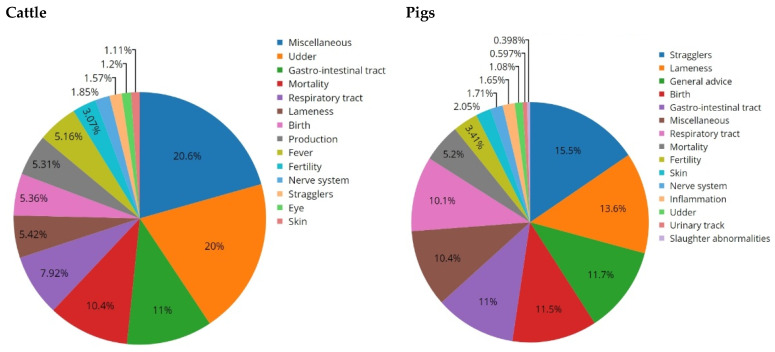
Frequency of reported clinical signs to the veterinary helpdesk for cattle and pigs in the Netherlands, per category as percentages of the total contact with reported clinical signs over the years 2015 to 2019 combined.

**Figure 3 animals-12-03344-f003:**
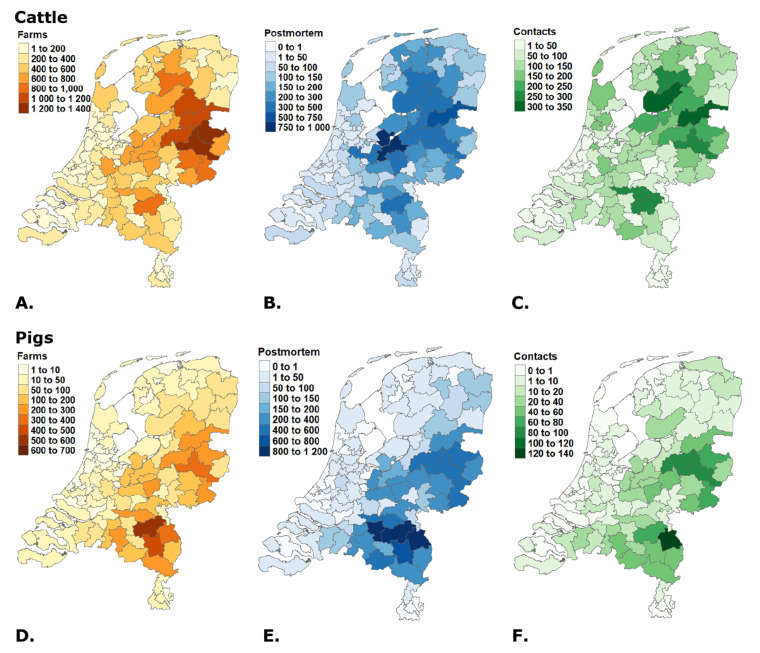
Distribution of farms (**A**,**D**), postmortem examinations (**B**,**E**) and helpdesk contacts (**C**,**F**) for cattle and pigs over the PC2 areas in the Netherlands from 2015 to 2019.

**Figure 4 animals-12-03344-f004:**
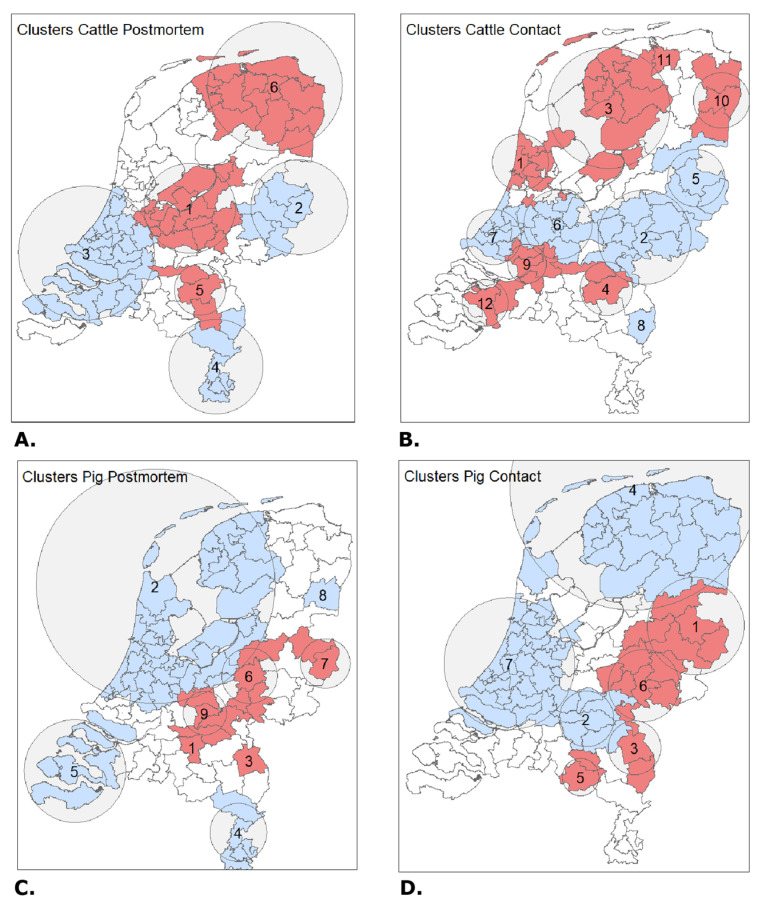
High (red) and low (blue) clusters for helpdesk contact and postmortem examination. (**A**) postmortem examination of cattle, (**B**) Helpdesk contacts cattle, (**C**) postmortem examination of pig, (**D**) helpdesk contacts pig. Numbers within clusters represent order of significance, all shown clusters *p* < 0.01.

**Figure 5 animals-12-03344-f005:**
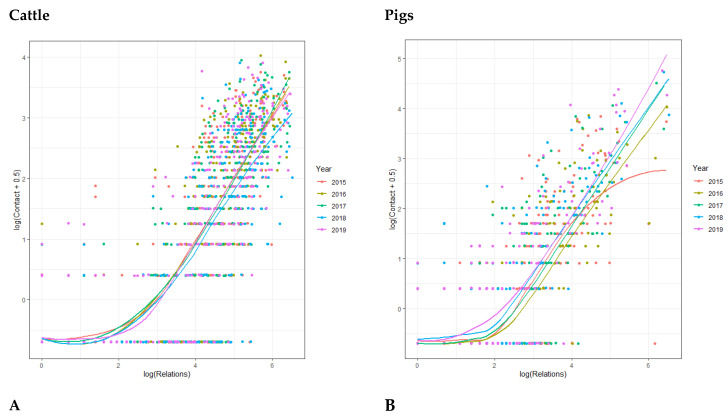
Scatterplot of the relation between contact with the veterinary helpdesk and number of registered relations of veterinary practices for cattle (**A**) and pigs (**B**).

**Table 1 animals-12-03344-t001:** Overview of inclusion criteria and used dataset per analysis of this study.

Analysis	Data		Inclusion Criteria		
	Helpdesk	Postmortem Examination	Herd *	Vet *	Registration of Clinical Signs
Clinical signs	√				√
Space-time analysis	√	√	√		
GLMM contact/postmortem examination	√	√	√		
GLMM veterinary practices	√			√	

* Location and ID was available.

**Table 2 animals-12-03344-t002:** SatScan settings used for the space-time analysis, different from default settings.

	Option	Setting
Analysis	Type of analysis	Space-time
	Scan for areas with	High and low Rates
	Max. spatial cluster size	15% of population at risk
	Min. temporal cluster size	2 years
	Max. temporal cluster size	4 years
Output	Print ASCII column headers	Yes
	Geographical output	All Yes
	Column Output Format	All ASCII Yes

**Table 3 animals-12-03344-t003:** Categorization of veterinary practices based on the number of farm relations.

Category	Cattle	Pigs
1	<15	<10
2	15–80	10–30
3	81–180	31–60
4	181–300	61–100
5	>300	>100

**Table 4 animals-12-03344-t004:** Descriptive statistics of veterinary helpdesk contacts and postmortem examinations for cattle in the Netherlands from 2015 to 2019.

	Number of Observation per Year (%)					
Contact	2015	2016	2017	2018	2019 ^1^	Total
Total observations	4107 (100)	4221 (100)	4035 (100)	3554 (100)	3698 (100)	19,615 (100)
Farm registered	1848 (45)	1836 (43)	1600 (40)	1654 (47)	2119 (57)	9057 (46)
Veterinary practice registered	2213 (54)	2327 (55)	2420 (60)	1946 (55)	2307 (62)	11,213 (57)
Farm and/or practice registered	3580 (87)	3682 (87)	3600 (89)	3144 (88)	3421 (93)	17,427 (89)
About clinical signs	2433 (59)	2649 (63)	2518 (62)	2096 (59)	2379 (64)	12,075 (62)
About possibly notifiable (Yes)	33 (0.8)	37 (0.8)	23 (0.6)	17 (0.5)	60 (1.6)	170 (0.8)
About possibly zoonotic risk (Yes)	438 (10.7)	466 (11.0)	485 (12.0)	446 (12.5)	269 (7.3)	2104 (10.7)
Postmortem examination						
Total submitted animals	2816 (100)	3003 (100)	2433 (100)	2507 (100)	2370 (100)	13,129 (100)
Farm ID	2812 (99.9)	2981 (99.3)	2431 (99.9)	2497 (99.6)	2369 (99.9)	13,090 (99.7)

^1^ New registration system.

**Table 5 animals-12-03344-t005:** Descriptive statistics of veterinary helpdesk contact and postmortem examinations for pigs in the Netherlands from 2015 to 2019.

	Number of Observation per Year (%)					
Contact	2015	2016	2017	2018	2019 ^1^	Total
Total observations	1262 (100)	1070 (100)	1185 (100)	1230 (100)	1300 (100)	6047 (100)
Farm registered	210 (17)	197 (18)	287 (24)	231 (19)	563 (43)	1488 (25)
Veterinary practice registered	844 (67)	746 (70)	838 (71)	917 (75)	1043 (80)	4388 (73)
Farm and/or practice registered	971 (77)	843 (79)	961 (81)	1000 (81)	1113 (86)	4888 (81)
About Clinical signs	507 (40)	473 (44)	554 (47)	567 (46)	671 (52)	2772 (46)
About Possibly notifiable (Yes)	2 (0.2)	1 (0.1)	1 (0.1)	5 (0.4)	13 (1.0)	22 (0.4)
About Possibly Zoonotic risk (Yes)	147 (11.6)	149 (13.9)	128 (10.8)	165 (13.4)	70 (5.4)	659 (10.9)
Postmortem examination						
Total submitted animals	2574 (100)	2496 (100)	2511 (100)	2456 (100)	2263 (100)	12,300 (100)
Farm ID	2567 (99.7)	2491 (99.7)	2505 (99.7)	2445 (99.6)	2257 (99.7)	12,265 (99.7)

^1^ New registration system.

**Table 6 animals-12-03344-t006:** Descriptive statistics of the total number of farms and veterinary practices in the Netherlands and the total number of farms and veterinary practices that had at least one moment of contact or submitted at least one animal for postmortem examination, respectively, in the period between 2015 and 2019.

		Total Numbers (%)				
National		2015	2016	2017	2018	2019
	Cattle Farms	35,448	35,176	35,032	33,652	32,448
	Pig farms	7528	7669	6777	6735	6395
	Veterinary practices (Cattle)	390	418	429	402	372
	Veterinary practices (Pig)	326	334	301	291	282
	Cattle	4,192,154	4,206,465	4,431,962	4,712,458	4,683,415
Unique ID ^1^						
Cattle	Contact Farm	1247 (3.5)	1227 (3.5)	1114 (3.2)	1053 (3.1)	1346 (4.1)
	Contact veterinary practices	231 (59.2)	227 (54.3)	234 (54.5)	206 (51.2)	218 (58.6)
	Postmortem examinations	1792 (5.1)	1859 (5.2)	1558 (4.4)	1646 (4.9)	1644 (5.1)
Pig	Contact Farm	160 (2.1)	160 (2.1)	194 (2.8)	167 (2.4)	347 (5.4)
	Contact veterinary practices	111 (34.0)	98 (29.3)	95 (31.6)	100 (34)	102 (36.2)
	Postmortem examinations	779 (10.3)	768 (10.0)	733 (10.8)	710 (10.5)	612 (9.6)

^1^ Number of farms and veterinary practices with at least one moment of contact or submitted at least one animal for postmortem examination (% of total farms or veterinary practices).

**Table 7 animals-12-03344-t007:** Association of distance to GD facility and farm density with helpdesk contact and postmortem examination from 2015 and 2019 in the Netherlands.

Contact			β	SE	IRR	95% CI IRR
Cattle	Intercept		−6.13 *	0.21	0	0.00–0.00
	Distance (10 km)		0.07 *	0.02	1.07	1.02–1.12
	Farm density	Lowest	0.66 *	0.15	1.94	1.46–2.59
		Low	0.35 *	0.1	1.42	1.17–1.72
		Moderate	Ref.			
		High	−0.1	0.07	0.91	0.79–1.03
		Highest	−0.19	0.16	0.82	0.60–1.14
	Year	2015	Ref.			
		2016	0	0.03	1	0.94–1.07
		2017	−0.02	0.03	0.98	0.91–1.05
		2018	0.07	0.04	1.07	1.00–1.14
		2019	0.11 *	0.03	1.11	1.04–1.19
Pigs						
	Intercept		−5.34 *	0.16	0	0.00–0.01
	Distance (10 km)		0.05*	0.02	1.05	1.02–1.08
	Farm density	Lowest	1.30 *	0.14	3.67	2.77–4.81
		Low	0.2	0.11	1.22	0.98–1.52
		Moderate	Ref.			
		High	−0.31	0.17	0.73	0.53–1.02
		Highest	−0.87 *	0.22	0.42	0.28–0.64
	Year	2015	Ref.			
		2016	−0.08	0.1	0.92	0.76–1.12
		2017	0.17	0.09	1.19	0.99–1.43
		2018	0.11	0.1	1.11	0.92–1.35
		2019	0.26*	0.08	1.3	1.11–1.54
Postmortem examination			β	SE	IRR	95 CI% IRR
Cattle	Intercept		−6.08 *	0.2	0	0.00–0.00
	Distance (10 km)		0.06 *	0.02	1.07	1.02–1.11
	Farm density	Lowest	0.71 *	0.14	2.04	1.54–2.69
		Low	0.19 *	0.09	1.21	1.01–1.45
		Moderate	Ref.			
		High	−0.1	0.06	0.91	0.80–1.02
		Highest	−0.18	0.12	0.84	0.66–1.06
	Year	2015	Ref.			
		2016	0.01	0.03	1.01	0.96–1.07
		2017	−0.01	0.03	0.99	0.93–1.05
		2018	−0.01	0.03	0.99	0.93–1.04
		2019	−0.03	0.03	0.97	0.90–1.03
Pigs						
	Intercept		−4.47 *	0.17	0.01	0.01–0.02
	Distance (10 km)		0.06 *	0.02	1.06	1.02–1.09
	Farm density	Lowest	1.14 *	0.14	3.14	2.39–4.14
		Low	0.09	0.07	1.1	0.95–1.26
		Moderate	Ref.			
		High	−0.07	0.14	0.93	0.70–1.23
		Highest	−0.22	0.18	0.8	0.57–1.14
	Year	2015	Ref.			
		2016	−0.02	0.04	0.98	0.90–1.06
		2017	0.12 *	0.04	1.13	1.04–1.22
		2018	0.15 *	0.04	1.16	1.07–1.26
		2019	0.22 *	0.04	1.25	1.15–1.36

* Significant (*p* < 0.05).

**Table 8 animals-12-03344-t008:** Association between the number of relations of a veterinary practice to the number of contact to the veterinary helpdesk from 2015 to 2019 in the Netherlands.

Cattle			β	SE	IRR	95 CI% IRR
	Intercept		−2.64 *	0.07	0.07	0.06–0.08
	Relations per veterinary practice	<15	−0.65 *	0.14	0.52	0.40–0.68
		16–80	Ref			
		81–180	0.05	0.07	1.05	0.93–1.20
		181–300	−0.08	0.07	0.92	0.80–1.06
		300<	−0.34 *	0.08	0.71	0.61–0.83
	Year	2015	Ref			
		2016	−0.04	0.08	0.96	0.83–1.12
		2017	−0.02	0.08	0.98	0.84–1.14
		2018	−0.21 *	0.08	0.81	0.70–0.95
		2019	−0.01	0.08	0.99	0.85–1.15
Pigs						
	Intercept		−2.14 *	0.11	0.12	0.09–0.15
	Relations per veterinary practice	<10	−0.94 *	0.18	0.39	0.31–0.49
		11–30	Ref			
		31–60	0.31 *	0.14	1.37	1.04–1.80
		61–100	0.47 *	0.17	1.59	1.15–2.21
		100<	0.05	0.14	1.05	0.79–1.39
	Year	2015	Ref			
		2016	−0.33 *	0.14	0.72	0.55–0.95
		2017	−0.06	0.14	0.95	0.72–1.25
		2018	0.35 *	0.14	1.42	1.08–1.88
		2019	0.39 *	0.14	1.47	1.11–1.95

* Significant (*p* < 0.05).

## Data Availability

Data is only available on request at G.S.

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
