# Peer review of "Coverage and Representativeness of Passive Surveillance Components for Cattle and Swine in The Netherlands"

_animals, 2022, doi:10.3390/ani12233344_

Round 1

Reviewer 1 Report

The study aims to evaluate the representativeness and coverage of the Dutch national animal health surveillance system in the cattle and pig sector of passively collected data from the veterinary helpdesk and postmortem examination.  Such an initiative is crucial in avoiding major disease outbreaks in the country and with trade partners.

I made a few comments in the attached file, please check it.

Reviewer 2 Report

In this paper, the Authors propose, using their own words, ‘to evaluate the representativeness and coverage of AHSS in the Dutch cattle and pig sector of passively collected data of the veterinary helpdesk  and postmortem examination', in order to assess its capabilities of response in the case, for example, of an emerging disease spread.

The document is quite well written, and it is organized as follows: Simple Summary and Abstract, a first Section containing an Introduction, a second Section on Materials and Methods, a third Section on Results, a fourth section on Discussion, a fifth Section containing the Conclusions, and finally the References.

After a detailed review, I think that some questions need to be addressed by the Authors in order to obtain further clarification about several aspects in the document. Please, see my comments below.

Main concerns:

•  The goals of the study and its conclusions are not clearly stated in the Abstract.

•  Perhaps the title should be modified to better fit the goals and conclusions of the study.

•  Did the Authors search for previous studies in the same line as this one? In other countries?

•  Please rewrite the sentence in Lines 97-99 for better readability. Same for the sentence in Lines115-116.

•  Have the Authors considered that the removal of cases with incomplete information (Lines 164-165, for example) might contribute to some biases in the study?

•  Please, specify the versions of the software used, as well as the specifications of the hardware that was employed in the study.

•  I think the 'Conclusions' section contents are an extension of the discussion, instead of stating clearly the contributions of the work to the field of study, its limitations and possible future work.

Other comments:
•  Please check the use of commas after words such as 'although' (Lines 61, 108), 'data' (Line 127), 'chosen' (Line 211), 'contacts' (Line 299), 'pigs' (Line 443).
•  Use the appropriate format for 'km2' (Line 236).
•  In Table 5, please use commas for separating thousands groups as it was done in Table 4. See also Line 407.
•  Add an space in 'data(Figure' (Line 337).
•  Check line spacing in the label of Figure 1.
•  Consider replacing 'Where' with 'While' in Lines 304 and 390.
•  Consider replacing 'none-specific' with 'non-specific' in Line 427.
•  Consider replacing 'Pigs' with 'pigs' in Line 510.
